# Maternal Low-Grade Chronic Inflammation and Intrauterine Programming of Health and Disease

**DOI:** 10.3390/ijms22041732

**Published:** 2021-02-09

**Authors:** Francesca Parisi, Roberta Milazzo, Valeria M. Savasi, Irene Cetin

**Affiliations:** 1Department of Woman, Mother and Neonate, ‘V. Buzzi’ Children Hospital, ASST Fatebenefratelli Sacco, 20141 Milan, Italy; roberta.milazzo@unimi.it (R.M.); irene.cetin@unimi.it (I.C.); 2Department of Biomedical and Clinical Sciences, “Luigi Sacco”, University of Milan, 20157 Milan, Italy; valeria.savasi@unimi.it; 3Department of Woman, Mother and Neonate, ‘L. Sacco’ Hospital, ASST Fatebenefratelli Sacco, 20157 Milan, Italy

**Keywords:** maternal chronic low-grade inflammation, obesity, fetal inflammation, placental inflammation, fetal programming

## Abstract

Overweight and obesity during pregnancy have been associated with increased birth weight, childhood obesity, and noncommunicable diseases in the offspring, leading to a vicious transgenerational perpetuating of metabolic derangements. Key components in intrauterine developmental programming still remain to be identified. Obesity involves chronic low-grade systemic inflammation that, in addition to physiological adaptations to pregnancy, may potentially expand to the placental interface and lead to intrauterine derangements with a threshold effect. Animal models, where maternal inflammation is mimicked by single injections with lipopolysaccharide (LPS) resembling the obesity-induced immune profile, showed increased adiposity and impaired metabolic homeostasis in the offspring, similar to the phenotype observed after exposure to maternal obesity. Cytokine levels might be specifically important for the metabolic imprinting, as cytokines are transferable from maternal to fetal circulation and have the capability to modulate placental nutrient transfer. Maternal inflammation may induce metabolic reprogramming at several levels, starting from the periconceptional period with effects on the oocyte going through early stages of embryonic and placental development. Given the potential to reduce inflammation through inexpensive, widely available therapies, examinations of the impact of chronic inflammation on reproductive and pregnancy outcomes, as well as preventive interventions, are now needed.

## 1. Introduction

Despite the myth of transplant allografts raising the idea that systemic immune suppression is an essential feature of pregnancy, the last decades of research revealed a more complex fetal–maternal immune interaction, involving both local and systemic inflammation as a crucial component of healthy pregnancies [1,2]. Starting with local inflammation, the human maternal–fetal interface involves a high number of immune cells, including macrophages, regulatory T cells, natural killers, and dendritic cells, which coordinate critical events of implantation, placental development, and finally delivery [1,3,4]. As a demonstration, experimental studies on decidual inflammatory cell deletion showed detrimental effects on blastocyst implantation and placental development, eventually leading to pregnancy termination [5,6]. On the other hand, pregnancy is notoriously related to systemic immunomodulation, with the most accredited theory suggesting a shift from Th1 to Th2 immunity and suppression of CD4+, CD8+, and natural killer cell response during pregnancy. As a result, enhanced immune tolerance has been clearly shown in normal pregnancies, with increased regulatory T cell numbers in response to fetal (paternal) antigen exposure and a typically more severe course of viral diseases [3,7,8]. 

Nevertheless, despite the fact that maternal inflammation plays a pivotal role in normal pregnancies, several maternal stressors, including malnutrition, environmental insults, and other pathological conditions driving excessive (prolonged or high-grade) systemic inflammation, may disrupt intrauterine environment. This may possibly lead to fetoplacental adaptations that initially aim to maintain pregnancy and fetal maturation, but secondly may increase the susceptibility to future noncommunicable diseases. This review summarizes the following points: 1. evidence of maternal low-grade chronic inflammation related to pregestational obesity; 2. pathogenic mechanisms linking maternal systemic low-grade chronic inflammation to intrauterine development; and 3. associations with intrauterine programming of future health and disease in both animal and human models. 

## 2. Pathobiology of Maternal Obesity

Human obesity is exponentially expanding worldwide to almost pandemic proportions, with an estimated prevalence of 30–70% of overweight and 10–30% of obese adults in European countries [9]. Increased short-term and long-term morbidity and mortality have been extensively reported for both obese mothers and offsprings, with about 24% of any pregnancy complication attributable to maternal overweight/obesity and one third of large-for-gestational-age babies to excessive gestational weight gain [10,11]. These epidemiological data well explain the critical need of understanding the biological processes driving adverse events in obese pregnancies.

### 2.1. Maternal Systemic Low-Grade Inflammation

Obesity involves a massive expansion of the adipose tissue, which is currently known as a highly active metabolic tissue with endocrine functions. Starting with the demonstration of tumor necrosis factor (TNF) production in mouse adipose tissue, going through the strong association detected between circulating TNF and insulin resistance to the final discovery that TNF inhibitors and weight loss are able to reverse insulin resistance in the same model, the cause-effect link between obesity, chronic inflammation, and metabolic derangements was lastly defined [12]. In line with these results, obese animal models showed improved insulin sensitivity after genetic deletion of singular inflammatory molecules, thus further confirming the role of chronic inflammation in obesity-induced insulin resistance [13]. The concept of adipose tissue as an endocrine organ was further confirmed in humans by the discovery of several secretory molecules, including adipokines, cytokines, and chemokines, able to perform the following functions: 1. signalling in an autocrine, paracrine, and endocrine network; 2. contributing to systemic low-grade chronic inflammation and endothelial dysfunction; and 3. depicting the so-called metabolic syndrome [12,14,15,16]. In both human and animal obese models, a pivotal role has been given to the adipose tissue macrophage infiltration (local inflammation) in determining the vicious cycle of inflammatory molecule secretion, insulin resistance, endothelial dysfunction, and increased body fat and body mass index (BMI) [17,18]. In contrast to acute responses, systemic inflammation in obese nonpregnant individuals is a low-grade and chronically perpetuated response, associated with a reduced metabolic rate and triggered by excessive nutrient consumption [19]. 

Fewer and often controversial data are available for investigating maternal systemic inflammation in obese pregnant women, and physiological adaptations to pregnancy may additionally obscure the underlying obesity-related inflammation. In fact, due to the central role of inflammation in the pathophysiology of both obesity and pregnancy, it is conceivable that interactions between maternal adaptations and obesity-related inflammation may result in aberrant upregulation of inflammatory mediators, thus contributing to short- and long-term morbidity in obese pregnancies under a threshold effect [20]. In this scenario, maternal obesity has been characterized as a metabolic inflammation showing increased circulating proinflammatory cytokines and adipose tissue macrophage accumulation, both finally extending to the placental interface and leading to an intrauterine proinflammatory environment [21,22]. The combination of obesity and other low-grade inflammatory diseases, such as periodontal disease, could synergistically amplify the inflammatory and pro-oxidative status, which can result in an elevation of local and systemic inflammatory biomarkers [23]. Furthermore, maternal serum interleukin (IL)-6 concentrations have been positively associated with fetal growth, thus linking the maternal proinflammatory environment to intrauterine overgrowth in obese mothers [24]. However, despite increased serum cytokines, including leptin, C-reactive protein (CRP), IL-6, and ICAM-1, were shown in obese human pregnancies compared to in matched controls, several reports did not confirm this result for all inflammatory markers [25,26,27,28,29]. 

### 2.2. Placental Inflammation

As the pivotal mediator between maternal and fetal environments, the subsequent step necessary to investigate associations between maternal chronic inflammation and intrauterine development needs a careful evaluation of placental development into the obesogenic environment. Pregestational obesity has been generally associated with increased birth and placental weight, lower placental efficiency (feto-placental weight ratio), accelerated villi maturation, and placental inflammatory cell infiltration [30,31,32]. As expected, local macrophage infiltration and increased proinflammatory mediators, including IL-6, leptin, and TNF, were shown in obese placentas [27]. Furthermore, increased uterine Treg lymphocytes accumulation, NK cell activity, and TNF production have been reported in both high-fat-fed mice and human models, indicating that maternal obesity promotes local uterine inflammation and cytokine secretion, which further modulate cellular function [33,34]. Since placenta- and adipose tissue-derived cytokines are critical regulators of placental expression of nutrient transporters, this could represent the causal link between maternal obesity, systemic and local inflammation, and increased placental substrate transport, leading to intrauterine overgrowth [27,35,36]. In particular, leptin (increased in obese mothers) has been shown to promote placental lipolysis, increase system A amino acid transport activity and stimulate IL-6 and nitric oxide release in cultured human trophoblast cells, thus potentially explaining increased nutrient transport, fetal overgrowth, and the vicious cycle resulting in increased local inflammation and oxidative stress [37,38]. In contrast, adiponectine (reduced in obese mothers) was shown to decrease amino acid transporter expression and uptake in human trophoblast cells [39,40]. 

Obesity-driven inflammation has further shown associations with trophoblast development and function in animal models, where leptin and low-density lipoproteins are able to regulate trophoblast apoptosis, proliferation, and migration in culture [2]. Consistently, high-fat-fed animals showed impaired uterine vascular remodeling, placental angiogenic defects, poor decidualization, and smaller implantation site in obese early pregnancy [41,42]. Human studies confirmed the role of several adipokines (i.e., leptin) in regulating placental angiogenesis, protein synthesis, and growth, finally impacting placental function in obese mothers [43,44]. Transcriptomic analyses of human obese placentas confirmed an increased expression of genes related to lipid metabolism, angiogenesis, and hormone/cytokine activity, resulting in a lipotoxic placental environment characterized by decreased vasculogenesis, increased oxidative stress, and fetoplacental hypoxia [2,45]. Moreover, placental metabolome analysis of obese pregnancies has also recently shown different patterns of amino acid profiles and mitochondrial function, supporting a shift towards higher placental metabolism. These placentas also showed a specific a fatty acids profile suggesting a disruption of LC-PUFA biomagnification [46].

All these alterations, together with a resulting mitochondrial dysfunction and excessive reactive oxygen species (ROS) production, may determine a cascade of events that lead to placental dysfunction and impaired pregnancy outcomes in obese pregnancies [47]. As a predictable result, newborns of obese mothers have been recently shown to be more hypoxic, acidemic, and with increased oxidative markers compared to normal-weight pregnancies [32,48]. 

Interestingly, despite impaired placentation in early pregnancy, animal experiments showed increased placental and birth weight at term in obese mice compared to in control mice, thus suggesting a compensatory mechanism later in pregnancy [41,42,49]. The same model found a reversal of adipose tissue and liver macrophage infiltration in obese pregnant dams compared to the pregestational status, almost to signify a reversal of the obesity-induced inflammation accordingly to increased immune tolerance during pregnancy [50]. Accordingly, human studies have recently reported a lower-term placental IL-6 expression, macrophage infiltration, GLUT1 and SNAT1-2 expression, and leptin production in obese pregnancies compared to in lean controls, as to signify a possible placental compensatory adaptation to the maternal obesogenic environment [51]. 

Finally, despite a decline in antioxidant response, decreased oxidative stress and damages have been shown in obese nondiabetic placentas compared to in lean controls, mainly explained by the activation of a nitric oxide-induced alternative pathway as a protective mechanism [52]. Given these results, it is possible that the feto–maternal interface may show profound development, structure, and function alterations as a consequence of maternal obesity, but also that the same plastic and dynamic interface may “sense” the obesogenic milieu and gradually adapt itself along pregnancy, in order to preserve fetal development. In agreement with this hypothesis, several studies investigating umbilical concentrations of inflammatory molecules (i.e., leptin, IL-6, and TNF) showed decreased or unaffected concentrations in neonates from obese mothers compared to in matched controls, thus identifying the placenta as an adaptor able to protect the fetus and maintain pregnancy [53,54]. To make data interpretation and prediction models even more complicated, several data additionally underlined a sex specificity of feto-placental adaptations to the obesogenic environment, meaning that male and female fetuses might implement different strategies to cope with the same detrimental stimulus [55,56,57]. As an example, placental inflammation and TNF levels were reported to be elevated in female placentas only, suggesting different placental inflammatory responses to obesity according to fetal sex [58].

### 2.3. Fetal Inflammation

Maternal chronic low-grade and placental inflammation have been controversially linked to feto-neonatal inflammatory derangements. In particular, the maternal cytokine environment may be specifically important for fetal metabolic imprinting, as cytokines are transferable from maternal to fetal circulation and have the capability to modulate placental nutrient transfer [59]. Maternal cytokines can cross the placenta and mediate a dialogue between the embryo and maternal tissues, which impacts on implantation success, blastocyst development, and long-term metabolic phenotype through effects on uterine receptivity, epigenome, cellular stress response, and apoptosis [60]. Increased placental free fatty acids, cholesterol, and triglycerides transport, reported in maternal obesity [61], have been shown to upregulate inflammatory pathways in fetal tissue of obese ewes, thus suggesting an indirect transmission of the inflammatory environment to the fetal district (i.e., toll-like receptor 4 (TLR4), nuclear factor-*κ*B (NF-*κ*B)) [62]. Accordingly, human offspring from obese mothers showed higher circulating concentration of C-reactive protein [63,64]. Nevertheless, the maternal proinflammatory environment has not been unequivocally linked to fetal inflammation. As an example, controversial increases in TNF*α*, IL-1, and IL-6 have been detected in fetal liver, brain, and plasma after maternal lipopolysaccharide (LPS) injection, but the gestational age and timing of injection need to be considered as impacting placental transport capacity [65,66]. In line with these results, some reports detected unaffected umbilical vein cytokine concentrations in newborns from obese mothers [54]. Together with cytokines, animal and human models demonstrated that also maternal immune cells are capable to cross the placental interface, infiltrate fetal tissue and persist into adulthood, but studies evaluating transplacental passage in case of maternal chronic Th-1 inflammation obesity-related are strongly needed [59,67].

Figure 1 summarizes the vicious cycle linking maternal low-grade chronic inflammation to feto-placental derangements with short- and long-term adverse outcomes.

## 3. Maternal Obesity-Related Inflammation and Developmental Programming 

The epidemiological evidence of a strong association between maternal nutritional status in pregnancy and metabolic syndrome in the offspring later life represents a keystone in the last decades of research [68]. In this context, the developmental origins of health and disease (DOHaD) hypothesis settles the strong association between what happens during the first period of life during gamete, embryonic, fetal, and early infant phases- and subsequent health and disease status [69]. The mechanisms underpinning the developmental programming of metabolic diseases still remain matter of debate, but a general pivotal role has been given to intrauterine adaptations to nutritional challenges that, may be maladaptive in later life despite maximizing the immediate chance of survival. As a consequence, the degree of mismatch between pre- and postnatal environments would represent a major determinant of subsequent disease risk [70,71]. However, a no-answer question has repeatedly been stressed by researchers: how can opposite nutritional stressors (over- and undernutrition) lead to the same programmed phenotype of obesity, cardiovascular disease, and insulin resistance in the offspring? [72,73]. In this scenario, the described systemic and placental adaptations to the maternal environment may lead to abnormal intrauterine development and disease programming through suboptimal maternal diet and an acquired proinflammatory phenotype during critical periods of development. In fact, growing evidence supports that maternal obesity-related inflammation might program offspring appetite, gene expression, immunity, gut microbiota, and adipocyte function [22,74]. The following sections summarize evidence from animal and human models on this topic, with the latter trying to provide the biological mechanisms of fetal programming by obesity-related maternal inflammation. 

### 3.1. Evidence from Animal Models

As human studies involve difficult discrimination between genetic and environmental contributions to offspring disease status, numerous obesogenic animal models have been performed in order to detect causative associations between intrauterine exposure to obesity and offspring disease, as well as to define the effect magnitude according to the timing of the nutritional insult and the severity of maternal disease (Table 1). 

The most commonly used animal models of developmental programming include rodents, sheep, and nonhuman primates. Animal models of maternal obesity may involve maternal inflammation as following: 1. mimicked by injections of lipopolysaccharide (LPS, a gram-negative bacteria-derived endotoxin, capable of propagating a type 1-based immune response similar to the obesity-induced inflammation) or 2. diet-induced (high-fat diet, high-sugar diet, and “cafeteria diet” mimicking a complex western dietary pattern). In general, a diet-induced maternal obesity induced obesity, altered brain appetite, insulin and leptin resistance, hypertensive disorders, reduced pancreatic beta-cell function, hepatic steatosis, and nonalcoholic fatty liver disease in the offspring [94]. Animal models of LPS injection are then of particular interest in order to define to what extent these effects are related to the inflammatory counterpart of maternal obesity. LPS injection in pregnant animal models has been associated with increased adiposity, systemic arterial blood pressure, and appetite, with contemporary decreased insulin sensitivity in the offspring, thus mimicking the same metabolic derangements obtained in the offspring from obese mothers [76,77].

Unbalanced maternal nutrition, particularly if rich in fats, was shown to induce maternal inflammation, further transmitted to the offspring through the activation of several inflammatory pathways, including peroxisome proliferator-activated receptors (PPAR) and NF-*κ*B signaling [75,81]. In line with this result, maternal LPS-stimulated inflammation in rodents determined a proinflammatory macrophage phenotype and enhanced IL-1*β* production in adult progeny under immune challenge [84]. Moreover, a recent animal model of chronic LPS-induced maternal inflammation found sex-specific leukocyte glucocorticoid hypersensitivity and exaggerated inflammatory cytokine responses in two generations of progeny [91,92]. In a mouse model, despite a comparable birth weight, a maternal high-fat diet and normal diet were shown to be associated with inflammatory changes in the adipose tissue of the offspring, including increased chemokine and cytokine expression [85]. This suggests an effect of obesity on fetal body composition and inflammatory profile independent on intrauterine growth trajectory. Taken together these models, simulating the chronic inflammation peculiar of obese mothers suggests that a maladapted proinflammatory maternal phenotype may be transmitted to the intrauterine environment and perpetuated to subsequent generations even without additional insults [91]. 

Besides inflammation programming, animal models of maternal obesity have been developed in order to characterize the intrauterine programming of metabolic derangements, independently from other pre- and postnatal exposures. As already underlined, maternal obesity is strongly associated with fetal growth aberrations, from fetal growth restriction to fetal growth acceleration, i.e., macrosomia [95]. The ending point of these two opposite situations is a paradoxical common prenatal metabolic programming of insulin sensitive tissues (i.e., pancreas, adipocytes, and skeletal muscle), which further explains the common later phenotype of metabolic syndrome in adulthood. In this context, diet-induced animal models have clarified that maternal obesity is associated with the following changes in the offspring: 1. adipose tissue modifications: increased sex-dependent adipogenesis (greater in males) with adipocyte hypertrophy, local inflammation enhanced PPAR-*γ* expression (obesogenic gene strictly related to lipid metabolism, cytokine production, and adipogenesis), reduced *β* 2- and *β* 3-adrenoreceptor expression and increased cytokine mRNA expression [79,83,93]; 2. central modifications: programmed hyperphagia (greater in male progeny), independent on postnatal nutrition, as a result of altered hypothalamic energy sensors and epigenetic responses, leading to altered development, neuronal abnormal differentiation, and appetite dysregulation [86,88,96]; 3. liver modifications: hepatic inflammation, steatosis, and fibrosis, leading to increased risk of nonalcoholic fatty liver disease, increased triglyceride accumulation and lipogenesis, enhanced proinflammatory cytokine and serum insulin expression, and premature gluconeogenic gene activation with impaired carbohydrate metabolism [80,82,89]; 4. skeletal muscle modifications: enhanced macrophage infiltration, increased inflammatory properties with upregulation of *PPAR-*γ**, *TLR2–4*, *NF-*κ*B*, and *TNF*α** gene expression, intramuscular adipogenesis with adipocyte hypertrophy and hyperplasia, and reduced insulin receptor mRNA expression, together resulting in decreased muscular insulin sensitivity and functional impairment [77,81,87,90,97]. 

Controversial results on the effect of anti-inflammatory nutrients in pregnancy on reversing maternal obesity-related programming have been reported (i.e., omega 3 fatty acids, resveratrol, and curcumin) [98]. As an example, animal and in vitro studies on omega-3 fatty acid administration showed increased muscular GLUT1 expression and glucose oxidation, decreased macrophage infiltration and adipokine production in the adipose tissue, inhibition of liver lipogenesis and increased fat oxidation, and decreased PPAR signaling in insulin sensitive tissues, all reversing the intrauterine programmed effect of metabolic derangements, but human trials have been less consistent on this topic [78,99,100,101]. 

### 3.2. Evidence from Human Studies

Although strong associations were shown between intrauterine exposure to obesity and postnatal cardio-metabolic risk in models considering both prenatal and postnatal confounding factors, human data are more ambiguous and sparse [102,103]. More and more epidemiological evidence has suggested causal associations between environmental and nutritional insults during critical windows of development and permanent effects on adult individual’s health status [104]. In addition, evidence associating other maternal conditions, including preeclampsia and diabetes mellitus in pregnancy, with childhood obesity underlies maternal systemic low-grade chronic inflammation as the common denominator of metabolic programming [105,106,107].

The evidence associating the exposure to the Dutch famine during early gestation with offspring obesity, insulin resistance, and cardiovascular disease came first to lay the foundation of the metabolic programming theory [108,109,110]. In the wake of these results, several studies proposed maternal pregestational BMI and gestational weight gain as independent predictors of childhood weight and body composition, considering composite adjustment for genetic and other environmental confounders [107,111,112,113]. To corroborate this thesis, human studies further demonstrated a lower prevalence of obesity among children born after maternal gastric bypass surgery compared to in children born before surgery, thus suggesting a long-term metabolic benefit by reversing maternal obesity [114]. The timing of the nutritional insult needs to be also considered. As an example, only first-trimester gestational weight gain has been shown to affect childhood weight, suggesting a greater impact of early pregnancy on childhood programming [115]. 

In line with animal models, newborns from obese and diabetic mothers showed enhanced intrahepatic fat and adipogenesis potential of umbilical cord mesenchymal stem cells [116,117]. Furthermore, the transgenerational transmission of obesity has been confirmed in humans [118].

Epigenetic dysregulation of proinflammatory and metabolic-involved genes has been reported to mediate the association between early nutrition and adult disease susceptibility [119,120]. In particular, significant changes in DNA methylation have been detected in both umbilical cord and children blood of offspring exposed to maternal obesity (with or without gestational diabetes), thus suggesting permanent postnatal DNA methylation changes after intrauterine exposure to an obesogenic environment [121,122].

Interestingly, the obesogenic environment has been shown to impact on genes related to inflammation, oxidative stress, and lipid metabolism already at the oocyte level, thus potentially affecting not only fertility outcomes, but also subsequent embryonic development [123,124]. In fact, the so-called periconceptional period, including crucial biological processes of gamete maturation, fertilization, and early embryonic development, has been shown to be particularly critical for developmental programming of health and disease, as it potentially leads to larger phenotypic shifts on the offspring than later insults [125]. Higher concentrations of insulin, triglycerides, leptin, lactate, and proinflammatory cytokines have been reported in the ovarian tissue of obese women, further associated with altered ovarian metabolic functions and reduced embryo developmental potential after fertilization [126,127,128]. The relevance of the metabolic reprogramming of ovarian cells has been demonstrated by the evidence that obese oocyte donors negatively affect reproductive outcomes despite not carrying the pregnancy [129]. During the periconceptional period, maternal obesity may impact on the offspring lifespan phenotype by perturbating or inducing compensatory adaptations on oocyte quality and oviduct and uterine luminal nutrient composition, as well as by inducing epigenetic modifications during early stages of embryonic and placental development [128,130]. This is in line with the conclusions by Snider et al. who noticed how the negative impact of obesity on ovarian function and oocyte quality has linked to increased inflammation and oxidative stress [131].

On the other hand, opposite results showing maternal BMI as strongly associated as the paternal counterpart with the offspring BMI suggested that the long-term effect may be independent on the intrauterine environment [132]. Thus, precise mediators and mechanisms involved in the intergenerational transmission of obesity in humans are still a matter of debate. 

## 4. Conclusions

Maternal obesity may affect long-term outcomes in the offspring as a result of intrauterine derangements and compensatory adaptations to a proinflammatory environment. Precise mediators of the developmental programming of maternal obesity are extremely difficult to decipher in humans, due to complex feto-placental–maternal interactions, difficult adjustment for pre- and postnatal confounding, and additional sex-specific responses.

Nevertheless, establishing associations between obesity-related inflammation and developmental programming are crucial, also given the potential to reduce inflammation through inexpensive and widely available therapies. In this context, nutritional intervention remains a promising and cost-effective target to modulate the transgenerational transmission of metabolic diseases, including the potential effect of antioxidant treatment, such as coenzyme Q10 [133], melatonin [134,135], omega-3, and healthy complex dietary patterns that could be the future cornerstones of research and clinical activities. In conclusion, animal and human models compositely provide strong evidence for the developmental origins of obesity through systemic, placental and fetal intrauterine adaptations able to perpetuate into adulthood. Cost-effective intervention strategies are now needed to ameliorate the unfavorable effects of maternal obesity on next generations. 

## Figures and Tables

**Figure 1 ijms-22-01732-f001:**
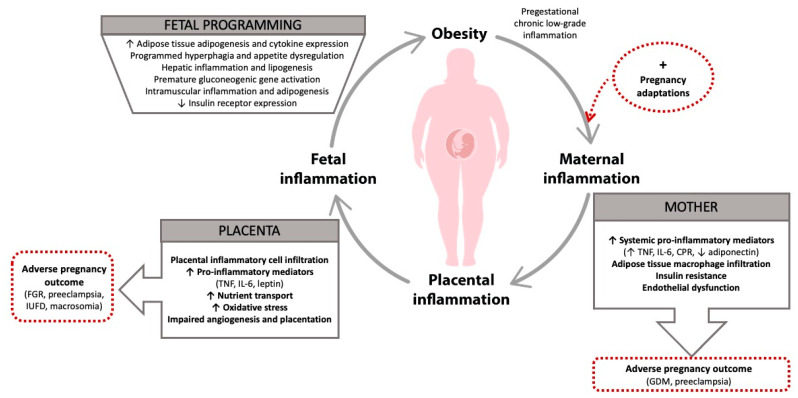
Transgenerational perpetuation of metabolic diseases through maternal low-grade chronic inflammation. The vicious cycle links maternal pregestational chronic low-grade inflammation to feto-placental inflammation with short- and long-term adverse outcomes. TNF: tumor necrosis factor; IL: interleukin; CRP: C-reactive protein; GDM: gestational diabetes mellitus; FGR: fetal growth restriction; IUFD: intra-uterine fetal demise. “↑” means increase; “↓” means decrease; “+” means additional.

**Table 1 ijms-22-01732-t001:** Animal models of obesity-related maternal inflammation and offspring outcomes. PPAR-*γ*: peroxisome proliferator-activated receptor gamma; TNF: tumor necrosis factor; TLR: toll-like receptor; NF-*κ*B: nuclear factor kappa-light-chain-enhancer of activated B cells; GLUT: glucose trasporter; G3PDH: glyceraldehyde-3-phosphate dehydrogenase; IL: interleukin; FGR: fetal growth restriction. “↑” means increase; “↓” means decrease.

	Animal Model	Inflammation Pathway	Offspring Outcomes	Maternal Outcomes
Hara et al. (2000) [75]	Rodent/HumanPro12Ala PPARgamma2 polymorphism	Diet-induced	Homozygous PPAR-*γ*-deficient embryos Spontaneous abortion Heterozygous PPAR-*γ*-deficient mice ↓ fat mass↑ leptin	Obese human mothers ↑ insulin sensitivity in presence of Ala12↓ Ala12 in the diabetic group
Nilsson et al. (2001) [76]	Rodent	LPS injection-induced	male offspring ↑ weight, adiposity, systemic arterial blood pressure and food intake↑ circulating leptin, 17beta-estradiol and progesterone↑ hippocampal glucocorticoid receptor expression↓ insulin sensitivity, corticosterone response to stress female offspring ↑ testosterone and baseline corticosterone levelsheart and adrenals enlargement	
Wei et al. (2007) [77]	Rodent	LPS injection-induced	↑ systemic arterial blood pressure, body weight, food intake, adipose tissue weight↑ circulating leptin	
Perez-Echarri et al. (2008) [78]	Rodent	Diet-inducedEicosapentaenoic (EPA) omega-3 fatty acid treatment	↑ TNF, IL-6 and haptoglobin in white adipose tissue↓ haptoglobin serum levels EPA treatment ↓ IL6 mRNA expression in white adipose tissue;Reversal of serum haptoglobin increase;Prevention of obesity-associated inflammation in adipose tissue	
Samuelsson et al. (2008) [79]	Rodent	Diet-induced	↑ body weight, blood pressure, adiposity with adipocyte hypertrophy, hyperphagia;↓ locomotor activity and skeletal muscle mass↓ adipocyte *β* 2- and *β* 3-adrenoreceptor and PPAR-*γ* expression;Systemic artery endothelial dysfunction and hypertension↑ fasting insulin and glucose levels	
Gregorio et al. (2010) [80]	Rodent	Diet-induced	Liver modifications:Insulin resistance and lower GLUT-2 expression↑ sterol regulatory element-binding protein-1c expression;Hepatic steatosis	
Yan et al. (2010) [81]	Sheep	Diet-induced	Skeletal muscle modifications:↑ PPAR-*γ*, TLR2–4, NF-*κ*B, and TNF*α* gene expression↑ intramuscolar adipogenesis and macrophage infiltration↑ circulating insulin concentrations↓ insulin receptor mRNA expression and insulin sensitivity	
Park et al. (2010) [82]	Rodent	Diet-induced	↑ tumor-promoting cytokines IL-6 and TNF which cause hepatic inflammation and activation of the oncogenic transcription factors → hepatocellular carcinoma development	
Rattanatray et al. (2010) [83]	Sheep	Diet-induced	↑ body fat mass in female offspring, reversible by maternal dietary restriction;No effect on PPAR-γ, G3PDH, lipoprotein lipase, leptin and adiponectin mRNA expression	
Kirsten et al. (2013) [84]	Rodent	LPS injection-induced	↑ IL-1β serum levels Male offspring autism-like behavior (impaired communication and socialization, repetitive/restricted behavior)	↑ maternal serum corticosterone levels, higher postimplantation loss
Murabayashi et al. (2013) [85]	Rodent	Diet-induced	= fetal weight↑ plasma glucose and insulin levelsadipocyte hypertrophy↑ adipocyte expression of chemokine receptor-2 and TNFα mRNA↓ adipocyte GLUT-4 expression	↑ bodyweight, glucose intolerance and insulin resistance
Desai et al. (2014) [86]	Rodent	Diet-induced (during pregnancy and/or lactation)	↑ adiposity↑ body weight only when overnutrition was prolonged during lactation;Hyperglycemia↑ systolic blood pressure↑ plasma corticosterone levels in case of maternal gestational overnutrition	↑ body fat and plasma corticosterone levels
Fink et al. (2014) [87]	RodentHuman	Diet-induced	Glucose intolerance Skeletal muscle modifications: ↑ macrophage infiltration by 76%;Dysregulated muscle inflammatory gene expression	
Desai et al. (2016) [88]	Rodent	Diet-induced	↑ adiposity despite unaffected body weight in males↓ energy sensors (DNA methylase)↑ appetite and ↓ satiety neuropeptides;Altered development, neuronal abnormal differentiation and appetite dysregulation in hypothalamus and adult arcuate nucleus	
Thompson et al. (2016) [89]	Rodent	Diet-induced	↑ hepatocyte proliferation and stellate cell activation; Hepatosteatosis↑ susceptibility to development of steatosis and rapid disease progression with sustained fibrotic phenotype	
Cadaret et al. (2018) [90]	Sheep	LPS injection-induced	Altered muscle metabolic capacity with ↓ glucose oxidation capacityFGR fetuses with –22% in body weight↓ β cell function	↑ circulating inflammatory cells
Adams et al.(2019; 2020) [91,92]	Rodent (F1 and F2 generations)	LPS injection-induced	Glucocorticoid hypersensitivity in F1 offspring with elevated corticosterone and increased leukocyte glucocorticoid receptor level, further transmitted to the F2 offspring without additional insults (>male offspring)↑ IL-1β cytokine responses in female offspring only	
Litzenburger et al. (2020) [93]	Rodent	Diet-induced	↑ white adipose tissue (female > male)↑ adipocyte sizeSex-dependent metabolic programming of white adipose tissue dysfunction with dysregulation of lipolytic, adipogenic and stemness-related markers

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
