# Peer review of "Maternal Low-Grade Chronic Inflammation and Intrauterine Programming of Health and Disease"

_ijms, 2021, doi:10.3390/ijms22041732_

Round 1

Reviewer 1 Report

The manuscript is devoted to the modern understanding of the effects of low-grade chronic inflammation in obese mothers on the programming
of a pathology in offspring.
The work is well structured, logical and understandable.
Paper provides links to both experimental animal models and human studies.
However, there are minor mistakes that need to be corrected:
line 105-108 The sentence is not clear. Especialy parts: "with several
reports not confirming this result" and "longitudinal trends". Need to rewrite this sentence.
line 117. What means "heterogeneous macrophage"? Explain please.

Have some misprint and fonts style different:
line 130 "mothers-"
line 151-152 fonts style
line 295-296 fonts style
line 299 "seruminsulin"
line 337 "disease-"
line 360 "on- oocyte"

Also figure illustrated not correctly. Some part of figure is not readable.
line 210-213 need to combine with headline of Figure 1.

Reviewer 2 Report

The authors present a comprehensive review of the effects of overweight and obesity on pregnancy outcomes in animals and women, including an overview of molecular mechanisms likely to be implicated. The paper is well written and clear. I recommend acceptance after minor revision along the following points.

Major points

  1. The words “overweight” and “obesity” should be included in the title. The current title does not reflect the article contents.
  2. The authors might suggest some unexpensive treatments that could be proposed to women of advanced age in whom applying dietary regimes might lead to further reduction of fertility due to ovarian aging. It appears that strong antioxidants, such as coenzyme Q10 (Xu et al. Reprod Biol Endocrinol 2018, 16, 29. doi: 10.1186/s12958-018-0343-0.) or melatonin (Tesarik and Mendoza-Tesarik. J Gynecol Women’s Health 2018, 12, 555829. doi: 10.19080/JGWH.2018.12.55582.)(Tamura et al. Int J Mol Sci 2020, 21, 1135. doi: 10.3390/ijms21031135.) might be of help. The latter is also known to modulate the immune system reducing uterine cell aggressivity towards the embryo.

Minor points

  1. The authors’ conclusions should be compared with those of another recent paper on the same subject (Snider and Wood, Reproduction 2019; 158, R79–R90, doi: 10.1530/REP -18-0583).

  1. In Figure 1, the central part is masking areas of the peripheral parts. This should be easy to correct.
